# Further Analysis of Outlier Detection with Deep Generative Models

**Ziyu Wang**[1,2]**, Bin Dai**[3]**, David Wipf**[4] **and Jun Zhu**[1,2]
[1] Dept. of Comp. Sci. & Tech., Institute for AI, BNRist Center,
Tsinghua-Bosch Joint ML Center, THBI Lab, Tsinghua University, Beijing, China
[2] Jiangsu Collaborative Innovation Center for Language Ability, Jiangsu Normal University
[3] Samsung Research China, Beijing, China
[4] AWS AI Lab, Shanghai, China
{wzy196,daib09physics,davidwipf}@gmail.com, dcszj@mail.tsinghua.edu.cn

## Abstract

The recent, counter-intuitive discovery that deep generative models (DGMs) can frequently assign a higher likelihood to outliers has implications for both outlier detection applications as well as our overall understanding of generative modeling. In this work, we present a possible explanation for this phenomenon, starting from the observation that a model's typical set and high-density region may not conincide. From this vantage point we propose a novel outlier test, the empirical success of which suggests that the failure of existing likelihood-based outlier tests does not necessarily imply that the corresponding generative model is uncalibrated. We also conduct additional experiments to help disentangle the impact of low-level texture versus high-level semantics in differentiating outliers. In aggregate, these results suggest that modifications to the standard evaluation practices and benchmarks commonly applied in the literature are needed.

## 1  Introduction

Outlier detection is an important problem in machine learning and data science. While it is natural to consider applying density estimates from expressive deep generative models (DGMs) to detect outliers, recent work has shown that certain DGMs, such as variational autoencoders (VAEs [1]) or flow-based models [2], often assign similar or higher likelihood to natural images with significantly different semantics than the inliers upon which the models were originally trained [3, 4]. For example, a model trained on CIFAR-10 may assign higher likelihood to SVHN images. This observation seemingly points to the infeasibility of directly applying DGMs to outlier detection problems. Moreover, it also casts doubt on the corresponding DGMs: One may justifiably ask whether these models are actually well-calibrated to the true underlying inlier distribution, and whether they capture the high-level semantics of real-world image data as opposed to merely learning low-level image statistics [3]. Building on these concerns, various diagnostics have been deployed to evaluate the calibration of newly proposed DGMs [5–9], or applied when revisiting older modeling practices [10].

As we will review in Section 5, many contemporary attempts have been made to understand this ostensibly paradoxical observation. Of particular interest is the argument from *typicality*. Samples from a high-dimensional distribution will often fall on a *typical set* with high probability, but the typical set itself does not necessarily have the highest probability density at any given point. Per this line of reasoning, to determine if a test sample is an outlier, we should check if it falls on the typical set of the inlier distribution rather than merely examining its likelihood under a given DGM. However, previous efforts to utilize similar ideas for outlier detection have not been consistently successful [3, 11]. Thus it is unclear whether the failure of the likelihood tests studied in [3] should be attributed

to the discrepancy between typical sets and high-density regions or instead, the miscalibration of the corresponding DGMs. The situation is further complicated by the recent discovery that certain energy-based models (EBMs) do actually assign lower likelihoods to these outliers [5, 6], even though we present experiments indicating that the probability density function (pdf) produced by these same models at out-of-distribution (OOD) locations can be inaccurate.

In this work we will attempt to at least partially disambiguate these unresolved findings. To this end, We first present an outlier test generalizing the idea of the typical set test. Our test is based on the observation that applying the typicality notion requires us to construct an independent and identically distributed (IID) sequence out of the inlier data, which may be too difficult given finite samples and imperfect models. For this reason, we turn to constructing sequences satisfying weaker criteria than IID, and utilizing existing tests from the time series literature to check for these properties. Under the evaluation settings in previous efforts applying DGMs to outlier detection, our test is found to work well, suggesting that the previously-observed failures of outlier tests based on the DGM likelihood should not be taken as unequivocal evidence of model miscalibration per se. We further support this claim by demonstrating that even the pdf from a simple multivariate Gaussian model can mimic the failure modes of DGMs.

Beyond these points, our experiments also reveal a non-trivial shortcoming of the existing outlier detection benchmarks. Specifically, we demonstrate that under current setups, inlier and outlier distributions can often be differentiated by a simple test using linear autocorrelation structures applied in the original image space. This implies that contrary to prior belief, these benchmarks do not necessarily evaluate the ability of DGMs to capture semantic information in the data, and thus alternative experimental designs should be considered for this purpose. We present new benchmarks that help to alleviate this problem.

The rest of the paper is organized as follows: In Section 2 we review the typicality argument and present our new outlier dectection test. We then evaluate this test under a range of settings in Section 3. Next, Section 4 examines the difficulty of estimating pdfs at OOD locations. And finally, we review related work in Section 5 and present concluding discussions in Section 6.

## 2 From Typicality to a White Noise Test

### 2.1 OOD Detection and the Typicality Argument

It is well-known that model likelihood can potentially be inappropriate for outlier detection, especially in high dimensions. For example, suppose the inliers follow the $d$-dimensional standard Gaussian distribution, $p_{\text{in}}(x) \propto \exp(-\|x\|_2^2/2)$, and the test sample is the origin. By concentration inequalities, with overwhelming probability an inlier sample will fall onto an annulus with radius $\sqrt{d}(1 \pm o(1))$, the typical set, and thus the test sample could conceivably be classified as outlier. Yet the (log) pdf of the test sample is higher than most inlier samples by $O(d)$. This indicates that the typical set does not necessarily coincide with regions of high density, and that to detect outliers we should consider checking if the input falls into the former set. We refer to such a test as the *typicality test*.

However, the typicality test is not directly applicable to general distributions, since it is difficult to generalize the notion of typical set beyond simple cases such as component-wise independent distributions, while maintaining a similar concentration property.[1] One appealing proposal that generalizes this idea is to fit a deep latent variable model (LVM) on the inlier dataset using a factorized prior, so that we can transform the inlier distribution back to the prior and invoke the typicality test in the latent space. This idea has been explored in [3], where the authors conclude that it is not effective. One possible explanation is that for such a test to work, we must accurately identify the LVM, which may be far more difficult than generating visually plausible samples, requiring a significantly larger sample size and/or better models. Overall, the idea of typicality has not yet been successfully applied to single-sample outlier detection for general inlier distributions.

## 2.2 A White Noise Test for Outlier Detection

As we focus on the high-dimensional case, it is natural to take a longitudinal view of data, and interpret a $d$-dimensional random variable $x$ as a sequence of $d$ random variables. From this perspective, the aforementioned LVM test essentially transforms $x$ to another sequence $T(x)$, so that when $x \sim p_{\text{in}}$, $T(x)$ is IID.[2] Given a new sample $x'$, the test evaluates whether $T(x')$ is still IID by checking the value of $\sum_{i=1}^{d} T_i(x')^2$. The statistical power of the test is supported by concentration properties.

Of course IID is a strong property characterizing the lack of any dependency structure in a sequence, and transforming a long sequence back to IID may be an unreasonable objective. Thus it is natural to consider alternative sequence mappings designed to achieve a weaker criteria, and then subsequently test for that criteria. In the time series literature, there are two such weaker possibilities: the martingale difference (MD) and white noise (WN). A sequence $x$ is said to be a MD sequence if $\mathbb{E}(x_t|x_{<t}) = 0$ for all $t$; $x$ is said to be WN if for all $s \neq t$, $\text{Cov}(x_t, x_s) = 0$, $\text{Var}(x_s) = 1$. It is thus clear that for sequences with zero mean and unit variance, MD is a weaker property than IID, and WN is weaker than MD.

While IID sequences are automatically MD and WN, we can also construct WN or MD sequences from inlier samples using residuals from autoregressive models per the following:

**Claim 2.1.** *Let $\tilde{R}_t(x) := x_t - \mathbb{E}_{p_{\text{in}}}(x_t|x_{<t})$ and $R(x) := \tilde{R}_t(x)/\sqrt{\text{Var}_{p_{\text{in}}}(\tilde{R}_t(x))}$; let $W_t(x) := x_t - \sum_{s=1}^{t-1} a_{ts} x_s$, where the lower triangular matrix $A = (a_{ts})$ is the inverse of the Cholesky factor of $\text{Cov}_{x \sim p_{\text{in}}}(x)$. Assume $\text{Var}_{x \sim p_{\text{in}}}(\tilde{R}_t) > 0$ for all $t$. Then when $x \sim p_{\text{in}}$, $\tilde{R}(x), R(x)$ are both MD, and $R(x), W(x)$ are both WN.*

The first claim above follows from definition. For the second, $R$ is WN because it is MD and has unit variance. Also, $W$ is WN since $\text{Cov}_{x \sim p_{\text{in}}}[W_t(x)] = I$.

The conditional expectations in $R$ can be estimated with deep autoregressive models. For convenience we choose to estimate them with existing autoregressive DGMs in literature (e.g. PixelCNN). However, even though we are fitting generative models, we only need to estimate the mean of the autoregressive distributions $\{p(x_t|x_{<t})\}$ accurately, as opposed to estimating the entire probability density function. For this reason, tests using $R$ should be more robust against estimation errors than tests based on model likelihood.

As testing for the MD property is difficult, we choose to test the weaker WN property. This can be implemented using the classical Box-Pierce test statistics [14]

$$Q_{\text{BP}} := d \sum_{l=1}^{L} \hat{\rho}_l^2, \tag{1}$$

where $\hat{\rho}_l$ is the $l$-lag autocorrelation estimate of a test sequence $(T_t(x))_{t=1}^{d}$. In practice, we can use either $W$ or $R$ as the test sequence, which are both WN when constructed from inliers. When $(T_t)$ has zero mean and unit variance, we have $\hat{\rho}_l = \frac{1}{d-l} \sum_{t=1}^{d-l} T_t T_{t+l}$. We consider a data point $x_{\text{test}}$ more likely to be outlier when $Q_{\text{BP}}(x_{\text{test}})$ is larger. Under the context of hypothesis testing where a binary decision (whether $x_{\text{test}}$ is an outlier) is needed, we can determine the threshold using the distribution of $Q_{\text{BP}}$ evaluated on inlier data.

In high dimensions, formally characterizing the power of a outlier test can be difficult; as illustrated in Section 2.1, it is difficult to even find a proper definition of outlier that is simultaneously practical. Nonetheless, the following remark provides some intuition on the power of our test, when the test sequence derived from outliers has non-zero autocorrelations. This is a natural assumption for image data, where the residual sequence from outlier data could contain more unexplained semantic information, which subsequently contributes to higher autocorrelation; see Appendix A for empirical verification and further discussion on this matter.

**Remark 2.1** (Connection with the concentration-of-measure phenomenon). *The power of the Box-Pierce test is supported by a concentration-of-measure phenomenon: When $\{T_t(x)\}$ is IID Gaussian,[3]*

$Q_{BP}$ will approximately follow a $\chi_L^2$ distribution [14], and $Q_{BP}/L$ will concentrate around $1$. On the other hand, if the null hypothesis does not hold and there exists a non-zero $\rho_l$, $Q_{BP}/L$ will be at least $d\rho_l^2/L$, which is much larger than $1$ when $d$ is large.

*It should be noted, however, that our test benefits from the concentration phenomenon in a different way comparing to the typicality test. As an example, consider the following outlier distribution: for $x \sim p_{\text{ood}}$, $(T_1(x), T_2(x))$ follow the uniform distribution on the circle centered at origin with radius $\sqrt{2}$, and $T_j(x) = T_{j-2}(x)$ for $j > 2$. Then $\frac{1}{d}\sum_{j=1}^d T_j^2(x) = 1$, and thus the typicality test cannot detect such outliers. In contrast, our test will always detect the lag-2 autocorrelation in $T$, and, as described above, reject the null hypothesis.*

### 2.3 Implementation Details

**Incorporating prior knowledge for image data:**   When applied to image data, the power of the proposed test can be improved by incorporating prior knowledge about outlier distributions. Specifically, as the test sequence $T(x)$ is obtained by stacking residuals of natural images, $\rho_l$ is likely small for the lags $l$ that do not align with fixed offsets along the two spatial dimensions. As the corresponding finite-sample estimates $\hat{\rho}_l$ are noisy (approximately normal), they constitute a source of independent noise that has a similar scale in both inlier and outlier data, and removing them from (1) will increase the gap between the distributions of the test statistics computed from inlier and outlier data, consequently improving the power of our test. For this reason, we modify (1) to only include lags that correspond to vertical autocorrelations in images. When the data sequence is obtained by stacking an image with channel-last layout (i.e., for $x_{3(W(i-1)+j)+c}$ refers to the $c$-th channel of the $(i,j)$ pixel of a $H \times W$ RGB image), we will only include lags that are multiples of $3W$. For empirical verifications and further discussion on this issue, see Appendix A.

**Testing on transformed data:**   Instead of fitting autoregressive models directly in the input space, we may also fit them on some transformed domain, and use the resulting residual for the WN test. Possible transformations include residuals from VAEs and lower-level latent variables from hierarchical generative models (e.g. VQ-VAE).[4] This can be particularly appealing for the test using $(W_t)$, as linear autoregressive models have limited capacity and cannot effectively remove nonlinear dependencies from data, yet the lack of dependency seems important for the Box-Pierce test, as suggested by Remark 2.1.

## 3   Evaluating the White Noise Test

In this section we evaluate the proposed test, with the goal of better understanding the previous findings in [3]. We consider three implementations of our white noise test, which use different sequences to compute the test statistics (1):

- the residual sequence $R$, estimated with autoregressive DGMs (denoted as **AR-DGM**);
- the residual sequence $W$ from a linear AR model, directly fitted on the input space (**Linear**);
- the sequence $W$ constructed from a linear model fitted on the space of VAE residuals (**VAE+linear**).

Note that both $R$ and $W$ can be viewed as constructed from generative models: for the sequence $W$, the corresponding model is a simple multivariate normal distribution. Therefore, we can always gain insights from comparing our test to other tests based on the corresponding generative model.

Code for the experiments is available at `https://github.com/thu-ml/ood-dgm`.

### 3.1   Evaluation on Standard Image Datasets

We first evaluate our white noise test following the setup in [3], where the outlier data comes from standard image datasets, and can be different from inlier data in terms of both low-level details (textures, etc) as well as high-level semantics. In Appendix B we present additional experiments under a similar setup, in which we compare with more baselines.

**Evaluation Setup:**   We use CIFAR-10, CelebA, and TinyImageNet images as inliers, and CIFAR-10, CelebA and SVHN images as outliers. All colored images are resized to $32 \times 32$ and center cropped when necessary. For deep autoregressive models, we choose PixelSNAIL [15] when the inlier dataset is TinyImageNet, and PixelCNN++ [16] otherwise. We use the pretrained unconditional models from the respective papers when possible; otherwise we train models using the setups from the paper.[5] For the VAE-based tests, we use an architecture similar to [17], and vary the latent dimension $n_z$ as it may have an influence on the likelihood-based outlier test. See Appendix C.1 for more details.

We compare our test (**WN**) with three baselines that have been suggested for generative-model-based outlier detection: a single-sided likelihood test (**LH**), a two-sided likelihood test (**LH-2S**), and, for the DGM-related tests, the likelihood-ratio test proposed in [18] (**LR**). The LH test classifies samples with lower likelihood as outliers. The LH-2S test classifies samples with model likelihood deviated from the inlier median as outliers. It can be viewed as testing if the input falls into the *weakly typical set* [12];[6] while there is no concentration guarantee in the case of general inlier distributions, it is natural to include such a baseline. The LR test is a competitive approach to single-sample OOD detection; it conducts a single-sided test using the statistics $\log \frac{p_{model}(x)}{p_{generic}(x)}$, where $p_{generic}$ refers to the distribution corresponding to some generic image compressor (e.g., PNG). Samples with a lower value of this statistics is considered outlier. The test is based on the assumption that outlier samples with a higher model likelihood may have inherently lower complexity, as measured by $\log p_{generic}$. The test statistics, having the form of a Bayes factor, and can also be viewed that comparing two competing hypotheses ($p_{model}$ and $p_{generic}$) without assuming either is true [20].

Table 1: AUROC values for the single-sample test, and average ranks within each group. **Boldface** indicates best results; underline indicates notable failures (AUC < 0.5).

| Inlier Dist. | | CIFAR-10 | | CelebA | | TinyImageNet | | Avg. |
|---|---|---|---|---|---|---|---|---|
| Outlier Dist. | | CelebA | SVHN | CIFAR-10 | SVHN | CIFAR-10 | SVHN | Rank |
| AR-DGM | LH | 0.88 | 0.16 | 0.82 | 0.15 | 0.28 | 0.05 | 3.67 |
| | LH-2S | 0.77 | 0.69 | 0.84 | 0.78 | 0.55 | 0.93 | 2.50 |
| | LR | 0.86 | 0.86 | 0.99 | 1.00 | 0.39 | 0.56 | 2.00 |
| | WN | 0.97 | 0.83 | 0.85 | 0.93 | 0.85 | 0.62 | **1.67** |
| VAE+Linear $n_z = 64$ | LH | 0.64 | 0.09 | 0.88 | 0.26 | 0.28 | 0.04 | 3.33 |
| | LH-2S | 0.47 | 0.81 | 0.85 | 0.69 | 0.51 | 0.87 | 3.00 |
| | LR | 0.39 | 0.90 | 0.98 | 0.99 | 0.64 | 0.91 | 1.83 |
| | WN | 0.64 | 0.67 | 0.93 | 0.99 | 0.92 | 0.99 | **1.50** |
| VAE+Linear $n_z = 512$ | LH | 0.76 | 0.04 | 0.81 | 0.09 | 0.19 | 0.01 | 3.33 |
| | LH-2S | 0.61 | 0.85 | 0.76 | 0.81 | 0.59 | 0.90 | 2.67 |
| | LR | 0.56 | 0.86 | 0.97 | 0.99 | 0.55 | 0.90 | 2.50 |
| | WN | 0.61 | 0.88 | 0.88 | 1.00 | 0.94 | 0.99 | **1.33** |
| Linear | LH | 0.77 | 0.02 | 0.72 | 0.03 | 0.11 | 0.00 | 2.50 |
| | LH-2S | 0.69 | 0.76 | 0.70 | 0.80 | 0.64 | 0.81 | 2.17 |
| | WN | 0.67 | 0.95 | 0.90 | 0.99 | 0.92 | 0.99 | **1.33** |

**Results and Discussion:**   We compare the distribution of the test statistics on the inlier test data and outlier datasets, and report the AUROC values. The results are shown in Table 1, where we observe that our WN proposal outperforms all the others in terms of the average ranking across testing conditions; see rightmost column. (We have deferred to Appendix C.1 the results of likelihood-based tests based on multivariate normal models fitted on VAE residuals, as those tests did not work well.)

Drilling further into details, we can see that our WN test generally outperforms the likelihood-based tests, and the single-side likelihood test exhibits pathological behaviors. This happens across all choices of generative models, including the simple Gaussian model corresponding to the linear test. Therefore, it is reasonable to doubt whether the previously observed failures of likelihood-based tests should be attributed to some undesirable properties of DGMs. Alternatively, those results may be better explained by the counter-intuitive properties of high-dimensional probability, as in Section 2.1.

Furthermore, the fact that we can always construct a principled test statistics out of generative models suggests that these models have in some sense calibrated behavior on such outliers. In other words, under these settings the models do know what they don't know. Our result is to be compared with the recent discovery that EBMs assign lower likelihood to outliers under this setting [5, 6], which naturally leads to the question of whether a calibrated DGM should always have a similar behavior. However, our findings are not necessarily inconsistent with theirs, as we explain in Section 4.

Comparison between our test and the LR test is more nuanced, as the latter is also competitive in many cases. Still, the LR test consistently produces a slightly higher average rank, and also has two cases of notable failures.

Finally, note that the simple linear generative model, especially when combined with the WN test, works well in most cases. This challenges the intuition that the inflexibility of a linear model would hamper outlier-detection performance, and has two-fold implications. First, these results indicate that the linear white-noise test could be useful in practice, as it is easy to implement, and does not have unexpected failures like the likelihood tests. Hence, it could be applied as a cheap, first test in a detection pipeline. And secondly, the success of the linear test shows that the current benchmarks leave a lot to be desired, since it implies that the differences between the inlier and outlier distributions being exploited for outlier detection are mostly low-level. Consequently, it remains unclear if these benchmarks are adequate for showcasing tests that are sensitive to semantic differences. Such a semantics-oriented evaluation is arguably more important for downstream applications. Moreover, it better reflects the ability of DGMs to learn high-level semantics from data, as was the intent of [3]. To address this issue, in the following subsection we conduct additional experiments that are more focused on semantics.

## 3.2 Semantics-Oriented Evaluation

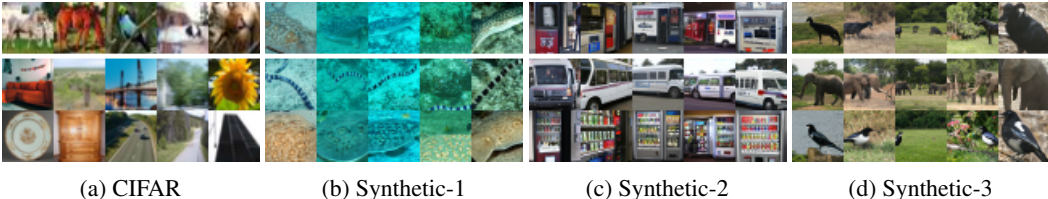

| (a) CIFAR | (b) Synthetic-1 | (c) Synthetic-2 | (d) Synthetic-3 |

Figure 1: Overview of inlier (top) and outlier (bottom) distributions used in Section 3.2.

In this section we evaluate the OOD tests in scenarios where the inlier and outlier distributions have different semantics, but the influence from background or textual differences is minimized. We consider two setups:

- **CIFAR**, in which we use CIFAR-10 images as inliers and a subset of CIFAR-100 as outliers. In this setup the inlier and outlier distributions have significantly different semantics, as we have removed from CIFAR-100 all classes that overlap with CIFAR-10, namely, non-insect creatures and vehicles. Furthermore, this setup also reduces textual differences contributed by inconsistent data collection processes; note that both CIFAR datasets have been created from the 80 Million Tiny Images dataset [21].

- **Synthetic**, in which we further reduce the background and textual differences between image classes by using synthesized images from BigGAN [22]. The outliers are class-conditional samples corresponding to two semantically different ImageNet classes; the inlier distribution is obtained by interpolating between these two classes using the GAN model. In this case, the semantic difference between inlier and outlier distributions is smaller, although in most cases it is still noticeable, as shown in Figure 1. We construct three benchmarks under this setting. Detailed settings and more sample images are postponed to Appendix C.2.

The results are summarized in Table 2, with full results for the synthetic experiments deferred to Appendix C.2. In the CIFAR setup, none of the tests that are based on the AR DGM or the vanilla Gaussian model works well, which is consistent with the common belief that these models cannot capture the high-level semantics. When using VAEs, the WN test works well. This experiment

Table 2: Results for the semantics-oriented experiments. Boldface indicates the best result.

| | CIFAR, AUROC↑ | | | | Synthetic, Avg. Rank↓ | | | |
| | LH | LH-2S | LR | WN | LH | LH-2S | LR | WN |
|---|---|---|---|---|---|---|---|---|
| AR-DGM | 0.49 | 0.57 | **0.61** | 0.58 | **2** | 3.5 | 2.5 | **2** |
| Linear | 0.56 | 0.59 | - | **0.60** | 2.33 | **1.67** | - | 2 |
| VAE+Linear, 64 | 0.51 | 0.55 | 0.64 | **0.84** | **1.67** | 3.33 | 2.67 | 2.33 |
| VAE+Linear, 512 | 0.59 | 0.58 | 0.73 | **0.80** | **2** | 3.67 | **2** | 2.33 |

reaffirms that DGMs such as VAEs are able to distinguish between distributions with significantly different semantics, even though they may assign similar likelihood to samples from both distributions.

However, as we move to the synthetic setup where the semantic difference is smaller but still evident, the outcome becomes quite different. The LH test performs much better, and our test no longer consistently outperforms the others. It is also interesting to note that the LR test does not work well on the second synthetic setup (see Appendix C.2), and completely fails to distinguish between inliers and outliers when using an autoregressive DGM. To understand this failure, we plot the distributions of model likelihood and test statistics in Appendix C.2. We can see that the outlier distribution has a slightly higher complexity as measured the generic image compressor, contrary to the assumption in [18] that the lower input complexity of outliers causes the failure of likelihood-based OOD test.

The difference in outcome between these experiments and Section 3.1 demonstrates the difficulty in developing a universally effective OOD test. It is thus possible that in the purely unsupervised setting we have investigated, OOD tests are best developed on a problem-dependent basis. Compared with Section 3.1, we can also see that the previous evaluation setups do not adequately evaluate the ability of each test to measure semantic differences. For this purpose, our approach may be more appropriate.[7]

## 4   On the Difficulty of Density Estimation in OOD Regions

While DGMs such as GANs, VAEs, autoregressive models, and flow-based models tend to assign higher likelihoods to certain OOD images, high-capacity energy-based models have been shown at times to have the opposite behavior [5, 6]. This observation naturally leads to the question of whether calibrated generative models trained on natural image datasets should always assign lower likelihood to such outliers. In this section, we argue that such a question is unlikely to have a clear-cut answer, by showing that given the relatively small sample size of typical image datasets compared to the high dimensionality of data, density estimation on OOD regions is intrinsically difficult, and even models such as EBMs can make mistakes.

Specifically, we train a PixelCNN++ and the high-capacity EBM in [5] on samples generated by a VAE. Since by design we have access to (lower bounds of) the true log probability density of the inlier distribution, we can check if a test model's density estimation in OOD regions is correct, simply by comparing it to the ground truth.

Our ground truth VAE has the same architecture as in Section 3, with $n_z = 64$; training is conducted on CIFAR-10. The DGMs to be tested are trained using 80000 samples from the VAE, under the same setup as in the original papers. See Appendix C.3 for details. We generate outliers by setting half of the latent code in the VAE to zero. Such outliers are likely to have a higher density under the ground truth model, per the reasoning from Section 2.1. Therefore, a DGM that correctly estimates the ground-truth data pdf should also assign higher likelihood to them.

The distributions of density estimates are shown in Figure 2. We can see that while both the EBM and PixelCNN++ models being tested assign a higher relative likelihood to the outliers (note that the absolute likelihoods between different models are not comparable because of different scaling and offset factors), the inlier and outlier density estimates from the EBM overlap significantly (middle plot) as compared to analogous overlap within the ground-truth VAE (left plot). Such behavior

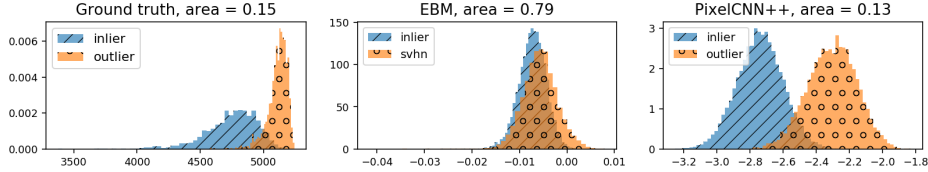

Figure 2: Distribution of log likelihood approximations from the ground-truth VAE (left), EBM (center), and PixelCNN++ (right). The intersection area of the two histograms is reported at the top.

may be attributed to the inductive bias of the EBM, which has a stronger influence than data on the estimated pdf in OOD regions given the relatively small sample size.

While we conjecture that VAEs or deep AR models can exhibit similar failures due to a different type of inductive bias, we cannot reverse the above experiment and train these models on EBM samples, as sampling from EBMs rely on ad hoc processes such as premature termination of MCMC chains [5, 6, 24]. Nonetheless, our experiment has demonstrated the intrinsic difficulty of density estimation in OOD regions under the finite-sample, high-dimensional setting. For this reason, it is difficult to draw a definitive conclusion as to whether real-world outliers should be assigned higher likelihoods, and alternative explanations, such as the typicality argument in Section 2, deserve more attention. The hardness of density estimation in OOD regions also suggests that OOD tests based on DGM likelihood should be used with caution, as is also suggested by the results in Section 3.1.

## 5 Related Work

Several works have explored the use of DGMs in outlier detection under settings similar to [3], some of which also provided possible explanations to the findings in [3]. For example, [11] presents a heuristic test using the Watanebe-Akaike Information Criterion; however, the efficacy of this test remains poorly understood. As another alternative, [25] proposes to compute the likelihood ratio between the inlier model and a background model, based on the intuition that background can be a confounding factor in the likelihood test. In Appendix B we present evaluations for the two tests, showing that they do not always work across all settings. In Section 3 we have introduced the work of [18], and demonstrated that its assumption does not always hold. In summary then, to date there has not been a comprehensive explanation of the peculiar behavior of generative models on semantically different outliers, although previous works can be illuminating and practically useful in certain scenarios.

For the general problem of high-dimensional outlier detection, methods have also been developed under different settings. For example, [19] proposes a typicality test assuming input contains a batch of IID samples, while [4] assumes a few outlier samples are available before testing. There is also work on outlier detection in supervised learning tasks, where auxiliary label information is available; see, e.g. [26–32].

Finally, it is worth mentioning the formulation of atypicality [33], as motivated by the possible mismatch between the typical set and the high-density regions. The atypicality test considers a test sequence to be OOD when there exists an alternative model leading to a smaller description length [34]. However, their choice to estimate $p(x_t|x_{<t})$ for *test* data $x$ becomes problematic when $x$ cannot be viewed as a stationary process, or with a large hypothesis space such as with DGMs.

## 6 Discussion

The recent discovery that DGMs may assign higher likelihood to natural image outliers casts into doubt the calibration of such models. In this work, we present a possible explanation based on an OOD test that generalizes the notion of typicality. In evaluations we have found that our test is effective under the previously used benchmarks, and that such peculiar behaviors of model likelihood are not restricted to DGMs. We have also demonstrated that certain DGMs cannot accurately estimate pdfs at OOD locations, even if at times they may correctly differentiate outliers. These findings suggest that it may be premature to judge the merits of a model by its (in)ability to assign lower likelihood to outliers.

Further investigation of the behavior of DGMs on outliers will undoubtedly continue to provide useful insights. However, our analyses suggest a change of practice in such investigations, such as considering alternatives to simply the model likelihood as our proposed test has exemplified. Likewise, the observation that a simple linear test performs well under current evaluation settings also suggests that care should be taken in the design and diversity of benchmark datasets, e.g., inclusion of at least some cases where low-level textures cannot be exclusively relied on.

And finally, from the perspective of unsupervised outlier detection, our experiments also revealed the intrinsic difficulty in designing universally effective tests. It is thus possible that future OOD tests are best developed on a problem-dependent basis, with prior knowledge of potential outlier distributions taken into account. [25] provides an example of such practice.

## Acknowledgement

Z.W. and J.Z. were supported by the National Key Research and Development Program of China (No. 2017YFA0700904), NSFC Projects (Nos. 61620106010, U19B2034, U1811461), Beijing Academy of Artificial Intelligence (BAAI), Tsinghua-Huawei Joint Research Program, a grant from Tsinghua Institute for Guo Qiang, Tiangong Institute for Intelligent Computing, and the NVIDIA NVAIL Program with GPU/DGX Acceleration. D.P.W. contributed to this project largely as an independent researcher prior to joining AWS.

## Broader Impact

This paper explores the nuances of applying DGMs to outlier detection, with the goal of understanding the limitations of current approaches as well as practical workarounds. From the perspective of fundamental research into existing machine learning and data mining techniques, we believe that this contribution realistically has little potential downside. Additionally, given the pernicious role that outliers play in numerous application domains, e.g., fraud, computer intrusion, etc., better preventative measures can certainly play a positive role. That being said, it is of course always possible to envision scenarios whereby an outlier detection system could inadvertently introduce bias that unfairly penalizes a marginalized group, e.g., in processing loan applications. Even so, it is our hope that the analysis herein could more plausibly be applied to exposing and mitigating such algorithmic biases.

## Footnotes

[1]While several papers have referred to the typical set for general distributions (e.g. a natural image distribution) which can be defined using the notion of weak typicality [12], we are only aware of concentration results for log-concave distributions [13], or for stationary ergodic processes [12]. Neither setting describes general distributions encountered in many practical applications.

[2]Note that such a transformation is possible as long as $p_{\text{in}}$ is absolutely continuous w.r.t. the Lebesgue measure; it does not require $x$ to represent truly temporal data.

[3] It is common to use the B-P test in the more general, non-IID case, so long as we are interested in alternative hypotheses where autocorrelation structure exist. Also recall that $\{T_t\}$ are residuals from an autoregressive model, so this condition is much weaker than requiring $x$ to be IID.

[4]Note this is different from testing with the sequence $R$, which is constructed from autoregressive models.

[5]This choice is made to maximize model capacity within the limit of computational resources we have.

[6] It can also be viewed as the single-sample version of [19].

[7] To balance the discussion, note that in some cases it may be desirable to have a benchmark outlier dataset with low-level differences, as such differences could be detrimental to down-stream applications. An example is the low-level differences of radiographs taken from different medical sites, which can influence diagnostics models [23]. Detection of such differences can be of practical interest in this context.

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
