[Supplementary Material]

# Appendix

## A    On the Assumptions and Efficacy of the White Noise Test

In this section we provide visualizations to better understand the statistical power of our test, and to verify the claims in Section 2.3.

We first plot samples of the residual sequence $R$ in Figure 3,[8] under varying choices of inlier and outlier distributions. We can see that $R$ constructed from outlier images generally include a higher proportion of unexplained semantic information: comparing the CelebA residual in Fig.3(a) (second column) where the model is trained on CIFAR-10, to Fig.3(b) (first column) where CelebA is inlier, we can see that the facial structure in CelebA residual is more evident when the model is trained on CIFAR-10. Similarly, comparing the CIFAR-10 residual from both models, we can see that the structure of the vehicle (e.g. front window and car frame) is more evident when the model is trained on CelebA. As the residual sequences constructed from outliers tend to have more natural image-like structures, they will also have stronger spatial autocorrelations, compared with residuals from inlier samples that should in principle be white noise.

Note that while the residual sequences constructed from inliers also contain unexplained semantic information, this is due to estimation error of the deep AR model, and should not happen should we have access to the ground truth model, as we have shown in Section 2.2. Moreover, the estimation error should have a small impact on the efficacy of the white noise test, as it is very easy to learn the correct linear autocorrelation structure of the inlier distribution, and thus the deviation of $R$ from WN is usually small, as we show in Figure 4 right.

(a) Inlier: CIFAR-10                                (b) Inlier: CelebA

Figure 3: Samples of normalized residual $R$ on different datasets, and the corresponding input images. The left 3 columns are generated from a PixelCNN++ trained on CIFAR-10; the right 3 columns corresponds to CelebA.

We now turn to the verification of our prior belief about the autocorrelation structure in $T(x_{\text{test}})$, when $x_{\text{test}}$ comes from the outlier distribution. Specifically, we plot the *average* ACFs on inlier and outlier data in Figure 4. We can see that the ACF estimates on outlier residuals peaks at lags that are multiples of 96, which corresponds to the vertical spatial autocorrelations in $32 \times 32 \times 3$ images. Moreover, on inlier and outlier distributions, the ACF estimates at other lags have approximately equal variances. When aggregated, these estimates will constitute a noticeable source of noise which reduces the gap between the distributions of inlier and outlier test statistics, and thus excluding them from the statistics will improve the power of the WN test.

Finally, we remark that it is also possible use spatial correlations directly in the construction of test statistics. However, our main focus in this work is to understand previous findings in generative outlier detection (instead of improving the state-of-the-art of OOD tests), and our choice to include only the vertical spatial autocorrelations is good enough for this purpose.

Figure 4: Averaged ACF estimates and their standard deviations (over sample images) on PixelCNN++ residuals. Left: residual generated from outlier (SVHN); right: from inlier (CIFAR-10) test set. Shaded area indicates the standard deviation of $\hat{\rho}_l$ *where the randomness is from the input data* $x$. Gray dashed line indicates the standard deviation of $\hat{\rho}_l$ *under the null hypothesis of IID residuals.*

## B   More Experiments on Standard Image Datasets

In this section we conduct additional experiments, and evaluate a variety of generative outlier detection methods under a common setting. As we will see, while several tests are in general more competitive than others, no single test achieves the best performance across all settings. This experiment strengthens our argument in the main text that unsupervised OOD tests should be developed on a problem-dependent basis.

**Evaluation Setup:**   We use CIFAR-10 as inlier data. For outliers we consider two setups. The first setup is taken from [18], and consists of 9 generic image datasets and 2 synthetic datasets, const and random; see Appendix A in [18] for details. The second setup controls for low-level differences by using the CIFAR-100 subset constructed in Section 3.2. The tests to be evaluated include those considered in Section 3.1, as well as the WAIC test [11] and the background likelihood ratio (BLR) test [25]. We base these tests on two DGMs: the VAE-512 model used in Section 3.1, and a smaller-capacity PixelCNN++ model as in [25].[9] For the BLR test, a noise level of the background model needs to be determined. Following the recommendations of the authors, we search for the optimal parameter in the range of $\{0.1, 0.2, 0.3\}$ using the grayscaled CIFAR-10 dataset as outlier. We found the optimal noise level to be $0.1$, which is consistent with [25].

**Results:**   Results are shown in Table 3-4. When using VAEs, neither of the newly added baselines are very competitive, suggesting that these methods are more prone to model misspecification. Notably, the WAIC test does not work with SVHN as outlier. This is also observed in [25, 19] using different generative models (autoregressive and flow-based models, respectively). For this reason we drop it in the PixelCNN++ experiment.

When we switch to PixelCNN++, the BLR test performs much better under the setting of [18]. However, in either case it does not work well with the subset-of-CIFAR-100 dataset, despite the dataset's clear semantic difference from the inlier dataset. Such results are not surprising since the difference in background or low-level details is much smaller for CIFAR-100 compared with the other datasets, as we have discussed in Section 3.2. Again, the difference in outcome between the two different settings demonstrates the difficulty of constructing universally effective OOD tests in the unsupervised setup.

## C   Experiment Details and Additional Results

### C.1   Details for Section 3.1

**Experiment Setup:**   For the AR-DGM experiments, we use the pretrained unconditional models from official repositories for CIFAR-10 and TinyImageNet. For CelebA we train a PixelCNN++ model using the authors' setup for unconditional CIFAR-10 generation. Both PixelCNN++ and

Table 3: Results for the Experiment in Appendix B, using VAE. Notable failures (with AUC$< 0.5$) are underlined.

| | LH | LH-2S | LR [18] | WAIC[11] | BLR[25] | WN |
|---|---|---|---|---|---|---|
| celeba | 0.76 | 0.61 | 0.57 | 0.64 | 0.22 | 0.62 |
| svhn | 0.04 | 0.85 | 0.86 | 0.14 | 0.96 | 0.88 |
| imagenet | 0.90 | 0.88 | 0.85 | 0.84 | 0.08 | 0.91 |
| facescrub | 0.65 | 0.47 | 0.52 | 0.55 | 0.33 | 0.60 |
| mnist | 0.28 | 0.42 | 1.00 | 0.68 | 0.81 | 0.83 |
| fashion | 0.45 | 0.36 | 1.00 | 0.64 | 0.61 | 0.83 |
| omniglot | 0.53 | 0.46 | 1.00 | 0.81 | 0.57 | 0.85 |
| trafficsign | 0.44 | 0.71 | 0.82 | 0.41 | 0.55 | 0.80 |
| random | 1.00 | 1.00 | 1.00 | 1.00 | 0.00 | 0.96 |
| const | 0.16 | 0.79 | 1.00 | 0.62 | 0.84 | 1.00 |
| avg. rank | 3.8 | 4.1 | **2.2** | 3.7 | 4.3 | **2.2** |
| cifar100' | 0.58 | 0.58 | **0.73** | 0.58 | 0.40 | **0.80** |
| (inlier test) | 0.45 | 0.47 | 0.43 | 0.63 | 0.44 | 0.49 |

Table 4: Results for the Experiment in Appendix B, using PixelCNN++. Notable failures (with AUC$< 0.5$) are underlined.

| | LH | LH-2S | LR[18] | BLR[25] | WN |
|---|---|---|---|---|---|
| imagenet | 0.86 | 0.82 | 0.88 | 0.92 | 0.84 |
| svhn | 0.11 | 0.79 | 0.80 | 0.79 | 0.86 |
| celeba32 | 0.81 | 0.64 | 0.75 | 0.89 | 0.97 |
| mnist | 0.00 | 1.00 | 1.00 | 0.91 | 0.98 |
| fashion | 0.00 | 1.00 | 0.97 | 0.82 | 0.96 |
| omniglot | 0.00 | 1.00 | 1.00 | 0.98 | 0.93 |
| facescrub | 0.80 | 0.69 | 0.82 | 0.93 | 0.82 |
| trafficsign | 0.55 | 0.59 | 0.90 | 0.90 | 0.77 |
| random | 1.00 | 1.00 | 1.00 | 1.00 | 1.00 |
| const | 0.09 | 0.87 | 1.00 | 0.04 | 1.00 |
| avg. rank | 4.44 | 3.22 | **1.89** | 2.78 | **2.56** |
| cifar100' | 0.50 | 0.57 | **0.63** | 0.45 | **0.58** |
| (inlier test) | 0.51 | 0.50 | 0.51 | 0.52 | 0.51 |

PixelSNAIL use the discretized mixture-of-logistics (DMOL) likelihood parameterization. To calculate its expectation, we first calculate the expectation of the continuous mixture of logistics distribution, and then *clip the result to the range of* $[0, 1]$. This is needed because the definition of the DMOL likelihood include a similar truncation [16]: extra probability mass for the interval $(1, +\infty)$ (or $(-\infty, 0)$) are assigned to the discretization bin $[1 - 1/256, 1]$ (or $[0, 1/256]$, respectively), so that the distribution is always supported on $[0, 1]$.

For the VAE experiments, we use the discretized logistics likelihood as the observation model. The network architecture is adapted from [17]; we vary the capacity of the model by increasing the number of filters in convolutional layers by $k$ times, where $k$ may be in $\{1, 2, 4, 8\}$. We train for at most $8 \times 10^5$ iterations using a learning rate of $10^{-4}$, and perform early stopping based on the validation ELBO. We choose $k$ to maximize validation ELBO. This leads to $k = 1$ for CIFAR-10, 4 for CelebA and 8 for TinyImageNet. This step is needed, because when $k$ is further increased, the reconstruction error will start to have different distributions between training and held-out set. Such a difference would be undesirable for all tests, as they will start to find false differences between the inlier training set and the test set. Note that this difference is not due to overfitting, as we have performed early stopping based on validation ELBO; instead, it is simply due to the fact that the model is exposed to training samples and not validation samples, and the gap appears very early in training. We use ELBO to approximate model likelihood in likelihood-related tests. The discrepancy between ELBO and true model likelihood is likely to have little impact on test performance, since we have also experimented with IWAE$_{100}$ which led to very similar results.

We compare the distributions of the test statistics evaluated on the inlier test set and outlier test set, and report the AUROC value. We verified that the four tests used in this section do not falsely distinguish between inlier training samples and test samples: the AUROC value for such a comparison is always in the range of $(0.42, 0.53)$. For outlier datasets with more than 50000 test samples, we sub-sample 50000 images for evaluation. Using the formula in [35], we can thus show that the maximum possible 95% confidence interval for the AUROC values is $\pm 0.011$. For a description of the four datasets used in this section, please refer to, e.g., Table 3 in [18].

**Choice of $L$ and Sensitivity:**   For our test, we use $L = 1200$ when computing the Box-Pierce statistics (1). This is because while in principle we should include all lags that are known *a priori* to be informative, in practice we only have $d - l$ samples to estimate $\hat{\rho}_l$, so the most distant lags can be difficult to estimate. Nonetheless, the impact of $L$ on the test outcome is relatively small: as is shown in Figure 5, using different $L$ does not lead to qualitatively different outcome. We also note that our purpose in the experiments is not to build new state-of-the-art in OOD detection, but is to use the proposed test to validate our explanation to previous findings. Still, if it is desirable to further improve the performance of the test, we can consider tuning $L$ on "validation outlier datasets" that is known *a priori* to be similar to the outliers that will be encountered in practice, as is done in e.g. [25].

Figure 5: Sensitivity to the maximum number of lags $L$ of the proposed WN test using AR-DGM. Inlier is CIFAR-10.

**Results for the Normal Likelihood Test on VAE Residuals:**   In Table 5 we present results for the likelihood tests using a multivariate normal model fitted on VAE residual, denoted with a prefix of "LN". We also consider both single-side and two-side tests. Overall the performance is similar to DGM likelihood, and the single-side likelihood test still manifests catastrophic failures.

Table 5: Full results for the VAE-related experiment in Section 3.1.

| Inlier Dist. | CIFAR-10 | | CelebA | | TinyImageNet | | Rank |
|---|---|---|---|---|---|---|---|
| Outlier Dist. | CelebA | SVHN | CIFAR-10 | SVHN | CIFAR-10 | SVHN | |
| **VAE-64** DGM-LH | 0.64 | 0.09 | 0.88 | 0.26 | 0.28 | 0.04 | 4.50 |
| DGM-LH-2S | 0.47 | 0.81 | 0.85 | 0.69 | 0.51 | 0.87 | 3.67 |
| LN-LH | 0.98 | 0.10 | 0.72 | 0.09 | 0.08 | 0.00 | 4.83 |
| LN-LH2S | 0.98 | 0.69 | 0.67 | 0.74 | 0.68 | 0.80 | 3.17 |
| LR | 0.39 | 0.90 | 0.98 | 0.99 | 0.64 | 0.91 | 2.33 |
| WN | 0.64 | 0.67 | 0.93 | 0.99 | 0.92 | 0.99 | **2.17** |
| **VAE-512** DGM-LH | 0.76 | 0.04 | 0.81 | 0.09 | 0.19 | 0.01 | 4.50 |
| DGM-LH-2S | 0.61 | 0.85 | 0.76 | 0.81 | 0.58 | 0.90 | 3.17 |
| LN-LH | 0.95 | 0.07 | 0.68 | 0.05 | 0.10 | 0.00 | 4.83 |
| LN-LH2S | 0.95 | 0.72 | 0.65 | 0.79 | 0.64 | 0.79 | 3.67 |
| LR | 0.56 | 0.86 | 0.97 | 0.99 | 0.55 | 0.90 | 3.00 |
| WN | 0.61 | 0.88 | 0.88 | 1.00 | 0.94 | 0.99 | **1.83** |

## C.2   Details for Section 3.2

**The CIFAR Experiment:**   We use the trained models from Section 3.1. We remove from CIFAR-100 the superclasses 1,2,9,12-17,19,20. For reference, the class names of CIFAR-10 and CIFAR-100 can be found in `https://www.cs.toronto.edu/~kriz/cifar.html`.

**The Synthetic Experiments:** We use a pretrained BigGAN model on ImageNet $128 \times 128$,[10] and down-sample the generated images to $32 \times 32$. To generate the outliers, recall the BigGAN generator takes as input a noise vector $z \in \mathbb{R}^{128}$ and the one-hot class encoding vector $c \in \mathbb{R}^{1000}$. Therefore, we interpolate between two classes $i$ and $j$ by setting $c_k = 0.5 \cdot \mathbf{1}_{k \in \{i,j\}}$. There are two tunable parameters in our generation process: the truncation parameter $\sigma$ that determines the truncated normal prior, and a crop parameter $\tau$. Before down-sampling the generated samples, we apply center-cropping to retain a proportion of $(1 - 2\tau)^2$ pixels, to reduce the amount of details lost in the down-sampling process. The classes and generation parameters used are listed in Table 6; they are hand-picked to ensure the background is similar in inlier and outlier classes. In each setup we generate 200000 samples and use 80% for training.

The VAEs are trained using the same setting as in Section 3.1. For PixelCNN++ we use the hyperparameters of the unconditional CIFAR-10 experiment in the original paper. As the synthetic datasets contain more samples, we train for 80 epochs.

The full AUROC values for the synthetic experiments are shown in Table 7. We plot the distributions of various statistics related to the LR tests using AR-DGM in the second synthetic experiment in Figure 6. We also plot additional inlier and outlier samples in Figure 7.

Table 6: Generation parameters for the synthetic experiment in Section 3.2.

| No. | Class 1 | Class 2 | $\sigma$ | $\tau$ |
|---|---|---|---|---|
| 1 | Sea Snake | Electric Ray | 0.8 | 0.25 |
| 2 | Bus | Vending Machine | 0.7 | 0.125 |
| 3 | Elephant | Magpie | 0.8 | 0.25 |

Table 7: AUROC scores for the synthetic experiments.

| | Synthetic 1 | | | | Synthetic 2 | | | | Synthetic 3 | | | |
| | LH | LH-2S | LR | WN | LH | LH-2S | LR | WN | LH | LH-2S | LR | WN |
|---|---|---|---|---|---|---|---|---|---|---|---|---|
| AR-DGM | 0.65 | 0.57 | 0.68 | 0.59 | 0.61 | 0.58 | 0.48 | 0.76 | 0.57 | 0.57 | 0.56 | 0.64 |
| Linear | 0.62 | 0.64 | - | 0.61 | 0.64 | 0.57 | - | 0.76 | 0.60 | 0.67 | - | 0.62 |
| VAE+Linear, 64 | 0.62 | 0.57 | 0.62 | 0.66 | 0.81 | 0.70 | 0.62 | 0.74 | 0.85 | 0.78 | 0.93 | 0.76 |
| VAE+Linear, 512 | 0.65 | 0.60 | 0.69 | 0.65 | 0.77 | 0.65 | 0.65 | 0.71 | 0.70 | 0.64 | 0.83 | 0.71 |

Figure 6: Distribution of various statistics related to the LR test using AR-DGM on the second synthetic experiment.

## C.3 Details for Section 4

The ground truth VAE has the same architecture as in Section 3.1, but with a continuous normal likelihood. We use $n_z = 64$. The VAE (log) likelihood is lower bounded by $\mathrm{IWAE}_{200}$. For EBM and PixelCNN++, we use the authors' hyperparameters and training setup for the unconditional CIFAR-10 experiments. After training, we verified that the distributions of energy values of training and held-out samples have small differences, so the models do not appear to overfit.

As the OOD test results in [5, 6] are obtained with conditional models, we perform the single-sided likelihood test with the unconditional model (trained on the real CIFAR-10 dataset) to check if its behavior on the SVHN dataset is similar to the conditional model. The AUROC value from the single-side likelihood test is 0.529, meaning that the EBM assigns similar or lower likelihood to

(a) CIFAR (left: inlier, right: outlier)

(b) Synthetic 1 (left: inlier, middle and right: outlier)

(c) Synthetic 2 (left: inlier, middle and right: outlier)

(d) Synthetic 3 (left: inlier, middle and right: outlier)

Figure 7: More sample images for the setups in Section 3.2.

SVHN compared with the inliers. This is still significantly different from the results using other generative models, justifying our use of an unconditional model.

## Footnotes

[8] With a slight abuse of notation, we use $R$ to refer to both the MD sequence constructed from true conditional expectation $\mathbb{E}_{p_{\text{in}}}(x_t|x_{<t})$, and the sequence constructed with DGM-based estimation to the conditional expectation.

[9]Using the standard hyperparameters in [16] results in the BLR test rejecting inlier test data as outlier with high confidence (AUROC> 0.9). As such a failure mode can be detected without access to outlier samples, we modify the model hyperparameters to follow [25] and train for 20 epochs. The BPD on inlier test set is 3.15.

[10]https://github.com/huggingface/pytorch-pretrained-BigGAN