[Reviews · NeurIPS 2020]

Review 1

Summary and Contributions: ----- Update ----- I have read the author response as well as the other reviews. I agree with some of the concerns raised by the other reviewers, but also do not find them to be significant to question the overall value and insights in this paper. I would still vote for accept, but lower my score to 7. ------------------ This work further analyzes the recently observed issue that Deep Generative Models (DGMs) regularly assign higher likelihood to out-of-distribution (OOD) samples/outliers. Based on the phenomenon that typical sets (regions of largest probability mass where samples likely fall into) must not coincide with density level sets (high-density/likelihood regions) in high dimensions, a novel white noise test for outlier detection is proposed. This test shows a marked improvement in detection performance over previous tests *using the same models* on common benchmarks (CIFAR-10, SVHN, CelebA, TinyImagNet in-/out-of-distribution combinations), thereby suggesting that DGMs are not necessarily uncalibrated, but rather that existing likelihood-based test might be improperly formulated/applied. Additional experiments on separating the effects of low-level textures vs. high-level semantic features are presented, followed by a discussion of current benchmarks as well as evaluation practices and the introduction of new semantic-focused benchmarks. The paper concludes with a discussion on the intrinsic difficulty of density estimation in a finite-sample, high-dimensional setting that also highlights model inductive biases (e.g. of recent EBMs) that also suggests future work is needed to better understand the issue.

Strengths: - The problem of high-likelihood outliers is a relevant and important problem from a scientific (understanding DGMs) but also practical perspective (trustworthy, robust models in applications). - The presented analysis is scientifically sound and presents new insights into this challenging problem, namely that the geometry of high-dimensional spaces must be taken into account for outlier testing and that the separation of low-level textures and high-level semantic features is crucial to the problem. - The proposed white noise test generalizes and improves over previous tests for typicality. - The suggestion and introduction of new semantic-oriented benchmarks extends and corrects shortcomings of the current testbed for OOD detection.

Weaknesses: - The analysis mainly focuses on model biases caused by learning objectives (e.g. likelihood or energy), but there are further inductive biases (e.g. network architecture) that very likely have an effect on the problem. This should be mentioned. - Important works of the related deep anomaly detection line of research [7, 5, 2, 3, 6, 1, 4] should be included which consider other objectives than likelihood/energy based on reconstruction, one-class classification, and self-supervision. ##### [1] L. Bergman and Y. Hoshen. Classification-based anomaly detection for general data. In ICLR, 2020. [2] I. Golan and R. El-Yaniv. Deep anomaly detection using geometric transformations. In NeurIPS, pages 9758–9769, 2018. [3] D. Hendrycks, M. Mazeika, S. Kadavath, and D. Song. Using self-supervised learning can improve model robustness and uncertainty. In NeurIPS, pages 15637–15648, 2019. [4] K. H. Kim, S. Shim, Y. Lim, J. Jeon, J. Choi, B. Kim, and A. S. Yoon. RaPP: Novelty detection with reconstruction along projection pathway. In ICLR, 2020. [5] L. Ruff, R. A. Vandermeulen, N. Görnitz, L. Deecke, S. A. Siddiqui, A. Binder, E. Müller, and M. Kloft. Deep one-class classification. In ICML, pages 4393–4402, 2018. [6] L. Ruff, R. A. Vandermeulen, N. Görnitz, A. Binder, E. Müller, K.-R. Müller, and M. Kloft. Deep semi-supervised anomaly detection. In ICLR, 2020. [7] S. Zhai, Y. Cheng, W. Lu, and Z. Zhang. Deep structured energy based models for anomaly detection. In ICML, volume 48, pages 1100–1109, 2016.

Correctness: - The reasoning and theoretical arguments are technically sound. - The empirical evaluation is scientifically rigorous.

Clarity: - The paper is well structured and written clearly. The overall exposition and presentation is excellent. - The careful, scientific tone and wording of this work is pleasantly refreshing in these times of tendencies towards verbally inflated contributions.

Relation to Prior Work: Overall, this work is well-placed into the existing literature and includes very recent works. However, the related deep anomaly detection line of research [7, 5, 2, 3, 6, 1, 4] that considers other objectives such as reconstruction, one-class classification, and self-supervision should be included to bridge these related lines.

Reproducibility: Yes

Additional Feedback: - The paper title is very generic.


Review 2

Summary and Contributions: This paper proposes a new outlier testing method that does not rely on likelihoods. Previously, the outlier testing methods based on the likelihood values of deep generative models have been shown to give inaccurate results. The proposed method argue that this is not exactly because the likelihood is not calibrated in the models but because the high likelihoods can be atypical in high-dimensional spaces. On this account, the authors present a new method relying on martingale-difference testing and white-noise testing.

Strengths: The paper is well-written and well-organized in general. The proposing method is a new line of research for outlier testing up to my knowledge, and the experimental results on popular image datasets seem promising.

Weaknesses: The paper could have given a more detailed introduction regarding the statistical tests. In particular, since the data are not exactly temporal sequences but being treated as one in the testing, it would be nice if such logical gaps are discussed. Some of my questions are: - How the typicality test is related to the IID condition and the weaker conditions (MD, WN)? - If it were a latent variable model (LVM) endowing the latent space with IID Gaussians, checking such conditions seems plausible. But in case of auto-regressive models and considering its MD construction (R(x)), why do we expect the in-distribution sequence to be white noise and outliers to be not?

Correctness: The claims and methods seem correct. The experiments, however, only cover the image datasets; the claims would be more strengthened if datasets from other domains are tested.

Clarity: The paper is clearly written in general, but there are several parts (mentioned in the weaknesses) that can be improved.

Relation to Prior Work: Yes. The prior work is discussed in the Sec. 5 and the comparative experiments have been conducted and reported.

Reproducibility: Yes

Additional Feedback:


Review 3

Summary and Contributions: The main contributions of this paper are: (1) It proposes a residual autocorrelation test for outlier detection and empirically reports that its performance surpasses those of likelihood-based methods on natural images (2) The authors show that the standard image datasets can be successfully distinguished using a simple linear model and proposes a new semantic-oriented evaluation where capturing high-level semantics is more important for OOD detection (3) It confirms that density estimation in OOD regions is hard for both deep generative models and energy-based models and suggests that current OOD evaluation protocols be revised or be developed on a problem-dependent basis.

Strengths: The proposed residual autocorrelation test is empirical powerful for standard image datasets. Moreover, it reveals that low textual information is often enough to solve the task, casting doubt on using the benchmark for DGM-based OOD detection methods. Also, the paper presents a new semantic-oriented OOD evaluation where textual or background difference is minimized and reveals that DGMs are useful for detecting different semantics when combined with the proposed white noise test. The empirical analyses show the difficulty of developing a universally effective OOD test, suggesting that future OOD tests be better configured in problem-dependent ways.

Weaknesses: One main question is: what are the potential differences between the proposed measure and the standard typicality test? For example, typicality is defined in terms of likelihoods where the white noise is defined as a lack of autocorrelations. What are the implications of this discrepancy for OOD detection? For instance, typicality requires a stronger IID assumption but that does not necessarily mean that typicality test cannot work in practice. (The concentration result for white noise test also relies on the IID assumption) What are the results if we apply typicality test on the residuals or on the transformed latent variables of DGMs? (As far as I know, Salisnick, et al. 2019 have not studied this matter in detail) It would be desirable for authors to discuss these matters in paper in more detail. In addition, though the white noise test is empirically promising, it relies on the assumption that DGMs will successfully remove autocorrelation structures from their residuals for inlier samples. However, the authors do provide some empirical evidence that there autocorrelation is scarce between the residuals of deep AR models in Appendix A.

Correctness: The claims and methods are reasonable and sound.

Clarity: The paper is clear overall.

Relation to Prior Work: While many prior works have explored the use of likelihood-based methods incorporating DGMs for OOD detection, this work is distinctive for that it proposes to test for autocorrelation structures in the residuals of DGMs instead of relying on likelihoods. In addition, it introduces a semantic-oriented evaluation where capturing higher-level semantics is more important in contrast to the standard image datasets which can be distinguished even with very low-level information.

Reproducibility: No

Additional Feedback: Typos - Line 251, "as measured" -> "as measured by"


Review 4

Summary and Contributions: The paper proposes a statistical test for detecting outliers, which performs much better than simple log likelihood test. The authors also explain why the log likelihood test can fail, and point to problems with designing a universal outlier detection test.

Strengths: The method is well motivated and clearly explained. The empirical tests are thorough. The paper is a valuable contribution to an actively researched area, and the method presented can have many important practical and scientific applications.

Weaknesses: Line 95: "While IID sequences are automatically MD and WN" is slightly inaccurate, as IID sequence may have E[x] <> 0 or Var[x] <> 1. Table 1 should report sample sizes used and confidence intervals for the AUROC values (see e.g. https://ncss-wpengine.netdna-ssl.com/wp-content/themes/ncss/pdf/Procedures/PASS/Confidence_Intervals_for_the_Area_Under_an_ROC_Curve.pdf).

Correctness: Overall yes. I wish the Authors commented more on the fact that their method depends on a deep autoregressive model (e.g. PixelCNN). How much would the results change if they used a different model?

Clarity: Yes, apart from some minor problems with style, e.g. "Samples with a lower value of this statistics is considered outlier" or "The test statistics, having the form of a Bayes factor, and can also be viewed that comparing two competing hypotheses (pmodel and pgeneric) without assuming either is true [19]" on page 5. I like the fact that the Authors take the trouble to motivate their work and make the problem easy to understand. Conference and journal papers should not be cited as arXiv abstracts (e.g. Kingma and Welling, 2014 or Hendrycks, Mazeika and Dietterich, 2018). Titles of papers should be formatted properly ("Bayes" not "bayes", etc).

Relation to Prior Work: Yes.

Reproducibility: Yes

Additional Feedback: I would try using the CIFAR-10.1 as an outlier set against CIFAR-10 as the inlier set. There is also the thorny question of the method's dependence on the autoregressive model used. How much do the results depend on the model used? === EDIT AFTER FEEDBACK === The Authors have addressed some of my concerns explicitly, and hopefully other manuscript quality problems will be resolved in the camera-ready copy. I keep the score unchanged.

[Author Response · NeurIPS 2020]

1   We thank all reviewers for their constructive comments.

**To Reviewer 1**

**Q1.** *Other inductive biases likely have an effect*:  We agree it is possible and will revise Sec. 4 to better reflect this.

**Q2.** *References on non-generative approaches to outlier detection*:  Thanks, we will include them in a revision.

**To Reviewer 2**

**Q1.** *Image data are not exactly temporal sequences*:  Even though the data is not truly temporal, the conditional distributions $\{p(x_i|x_{<i})\}$ are still well-defined, since the joint distribution $p(\{x_i\}_{i=1}^d)$ is defined. Thus for a random variable (rv) in $\mathbb{R}^d$ following a certain joint distribution (e.g. $p_{inlier}$), we can always view it as a sequence of $d$ rvs, where the $i$-th rv is sampled from $p(x_i|x_{<i})$. Subsequently, we can reason about properties of the random sequence, such as IID, MD or WN. We will revise the text to clarify this.

**Q2.** *How the typicality test is related to IID, WN and MD*:  The typicality test is only effective on factorized inlier distributions. So to apply this test, we need to transform the input rv so that the corresponding distribution becomes factorized.[1]  Usually the transformation is designed s.t. the resulting rv is also componentwise IID. Such a rv in $\mathbb{R}^d$ can be viewed as a *sequence* of $d$ IID rvs. Thus the typicality test amounts to testing the IID condition on the transformed sequence $T(x)$, using the statistic $\frac{1}{d}\sum_{i=1}^d T_i^2(x)$. It is not related to MD or WN, which are utilized by our test.

**Q3.** *For general AR models, why do we expect the sequence $R(x)$ to be WN for in-distribution $x$ and not WN for OOD $x$*:  **(i)** We showed that $R(x)$ is always WN when $x \sim p_{inlier}$ in L101-102. Note that we follow the convention in the time series literature, and define WN sequences as uncorrelated sequences with zero mean and unit variance; this definition does not require the sequence to be IID, and is called "weak WN" in some fields. **(ii)** For OOD $x$, $T(x)$ not being WN is the alternative hypothesis we are interested in. This is because designing a universally effective test is very difficult, if not impossible, given the high dimensionality compared to the limited number of samples. Thus we have to restrict our attention to certain alternative hypotheses (i.e. certain kind of outliers). In this work, we are interested in a variety of natural image outliers which have previously led to confusions. We believe that our alternative hypothesis suits this purpose, as discussed in L121-123 and verified in Appendix A.

**Q4.** *Test datasets from other domains*:  In this work we are mainly interested in natural image outliers, which have previously caused confusion over the calibration of DGMs in literature (also see L21-25, L31-36 in our submission).

**To Reviewer 3**

**Q1.** *Differences between the proposed test and the typicality test, and implications in practice (e.g. what happens if we apply the typicality test on non-IID sequences)*:  **(i)** They have different assumptions (WN vs IID), and the test statistics are different. **(ii)** The difference in assumptions means that it can be more difficult to apply the typicality test in a principled way, see L77-80. Even if we consider a heuristic application of the typicality test, i.e. to use the test statistics $\frac{1}{d}\sum_i T_i^2(x)$ where $T(x)$ isn't necessarily IID for inlier $x$, the difference in test statistics means that our test can still be more effective, since it could identify anomalous autocorrelation structures in the (transformed) *outlier* distribution. For example, suppose $T_i(x)$ is WN for inlier $x$, while for outlier $x$, $(T_1(x), T_2(x))$ are uniformly sampled from a centered circle with radius $\sqrt{2}$, and for $i > 2$, $T_i(x) = T_{i-2}(x)$. Then for such outliers, $\frac{1}{d}\sum_i T_i^2(x) = 1$, and the typicality test will not be able to detect them; in contrast, our test will detect such outliers, since they have *auto*correlations; see Remark 2.1. The autocorrelation issue is relevant for natural image outliers, as discussed in L121-123 and Appendix A.

**Q2.** *Results of the typicality test when applied on residuals, or the transformed latents of DGMs*:  **(i)** Nalisnick et al (2019) tested it using a flow model (Fig 4(a) therein) and showed that it was not effective. **(ii)** Some methods in Sec 3.1 are equivalent to the typicality test on certain whitened residuals: the LH-2S test using linear models, and the LN-LH2S method in Appendix (Table 5) which works on VAE residuals. Both methods are clearly outperformed by the corresponding WN tests. **(iii)** We also experimented with the typicality test applied to AR-DGM residuals, using the setup of Sec 3.1. The results are similar, with the typicality test outperformed by ours in 5 out of 6 cases.

**To Reviewer 4**

**Q1.** *Sample size and CI in Table 1*:  For all methods we use the entire test set, except for AR-DGM for which we sample $5 \times 10^4$ images from the larger datasets. This leads to a minimum sample size of $10^4$ (CIFAR-10 test set). Using the formula R4 provided, we can show that the maximum possible 95% CI is $\pm 0.011$. We will include them in revision.

**Q2.** *How much does the result change using different AR models*:  While we can't train new models due to time constraints, the results in Appendix B are obtained using a smaller-capacity PixelCNN++, and our test still works well. Also note that our test works with a simple linear AR model. Based on these results, it seems reasonable to expect that the results will not change qualitatively as we switch to different models.

## Footnotes

[1]This is the approach described in Sec. 2.1. See also L179-181 and footnote 1 in the submission, which discussed an alternative weak typicality test; that test is known to be ineffective, likely due to the lack of any concentration guarantee.


[Meta-Review · NeurIPS 2020]

The paper investigates out-of-distribution behavior of deep generative models, specifically the counter intuitive results reported in prior work where deep generative models were shown to assign higher likelihood to out-of-distribution inputs. The authors propose a new white noise test (WN test), theoretically motivate the proposed test and show that it outperforms likelihood and likelihood ratios. The reviewers raised concerns about experimental setup (other datasets and models), WN assumption and connections to other related methods such as typicality test. This was a borderline paper. During the discussion, majority of the reviewers agreed that the author rebuttal addresses their major concerns except for R2. After reading the paper carefully, I lean towards accept as the paper presents an interesting idea. Even though there are some gaps in the current version, the proposed revisions should strengthen the paper. I recommend comparison to typicality test with likelihood (e.g. batch_size=1) as that would help better situate this work in the wider literature and readers understand which are the biggest contributors to the better performance. Minor suggestion: Previous work has already suggested mismatch between high likelihood regions and typical set as a possible explanation. The authors cite these papers in the related work at the end of the paper, while some of the closely related papers [25,29] should probably be cited earlier in the intro when discussing the typicality explanation.